# Sustainable and Reliable Information and Communication Technology for Resilient Smart Cities

Nikolay Tcholtchev * and Ina Schieferdecker 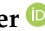

Fraunhofer Institute for Open Communication Systems (FOKUS), 10587 Berlin, Germany;
ina.schieferdecker@web.de
* Correspondence: nikolay.tcholtchev@fokus.fraunhofer.de; Tel.: +49-30-3463-7175

**Abstract:** Information and Communication Technology (ICT) is at the heart of the smart city approach, which constitutes the next level of cities' and communities' development across the globe. Thereby, ICT serves as the gluing component enabling different domains to interact with each other and facilitating the management and processing of vast amounts of data and information towards intelligently steering the cities' infrastructure and processes, engaging the citizens and facilitating new services and applications in various aspects of urban life—e.g., supply chains, mobility, transportation, energy, citizens' participation, public safety, interactions between citizens and the public administration, water management, parking and many other cases and domains. Hence, given the fundamental role of ICT in cities in the near future, it is of paramount importance to lay the ground for a sustainable and reliable ICT infrastructure, which can enable a city/community to respond in a resilient way to upcoming challenges, whilst increasing the quality of life for its citizens. A structured way of providing and maintaining an open and resilient ICT backbone for a city/community is constituted by the concept of an Open Urban Platform. Therefore, the current article presents the activities and developments necessary to achieve a resilient, standardized smart city, based on Open Urban Platforms (OUP) and the way these serve as a blueprint for each city/community towards the establishment of a sustainable and resilient ICT backbone.

**Keywords:** smart city; smart community; urban ICT; open urban platforms; sustainable cities; resilience; sustainability; reliability; quality assurance

## 1. Introduction

Smart cities and communities are at the forefront of innovation, research and development in modern societies. Subsequently, we use the term smart city as shorthand for smart city and community. As of 2020, more than 50% of the world population lives in cities [1], with predictions for a dramatic increase in the percentage of urban populations in the coming years [1]. Hence, there is an urgent need to optimize the processes within a city/community and to push for new eco-systems generating novel business and operational models for increasing the quality of life for citizens, whilst at the same time reducing costs and improving the city/community processes and operations. Thereby, ICT (Information and Communication Technologies) plays a vital role in enabling such eco-systems, given that they will emerge around the notion of data and information gathering, and making these data and information available across multiple domains towards the combination and exploitation of synergies amongst various aspects of urban business and everyday life. A smart city naturally emerges around an urban data platform, which consolidates various data sources across an urban eco-system. Indeed, the data sources can be versatile, including static data—e.g., governmental data, Open Data in general, and any sort of city data and information that do not constantly change in value/parameters—and dynamic data, e.g., continuous real-time data like sensor/Internet of Things (IoT) data, global positioning data, etc. In order to facilitate such a data driven approach, various

components, network segments, and computing nodes from different silos and domains need to interplay efficiently towards enabling a data driven smart city. This includes sensors generating and sending data over belonging gateways, which in turn forward the sensor data over telecom/Internet type of networks to data nodes at the network edge or data centers in the cloud. In addition, different services and applications—including mobile and embedded—utilize the data, analyze them and generate added value for end users, citizens, city decision makers and further stakeholders within an urban environment.

Along these considerations, it is very important to establish a holistic approach to the ICT in a smart city. Such a holistic approach should serve as a blueprint for establishing an urban ICT infrastructure and should enable the creation of dynamic and versatile eco-systems of large industry, small-mid size enterprises, start-ups, the public sector, open source communities, non-governmental organizations, applied research institutes, universities and city administrations. Such a holistic approach can be designed based on a reference architecture, which leans on other reference architectures coming from domains such as telecommunications (e.g., ISO/OSI (Open Systems Interconnection [OSI] Reference Model of the International Organization for Standardization [ISO]), TMN (Telecommunications Management Network defined by the International Telecommunication Union Telecommunication Standardization Sector) and others) or Internet data-based communication (e.g., TCP/IP (Transmission Control Protocol/Internet Protocol defined by the Internet Engineering Task Force [IETF]) and others). Such reference models have enabled the rapid growth of voice and data communication across the world and have led to one of the most impressive success stories of human kind—the Internet. Indeed, as the Internet and the telecommunication domain in general has proven to have developed in a sustainable and resilient way, we envision the same for the emerging ICT infrastructures within cities and communities that will serve as the backbone of future societies. There are various aspects that enabled the sustainability and resiliency (see also Section 3) of the Internet, including the following (1) reference architectures, the (2) implementation, quality assurance and certification procedures, the corresponding (3) standardization activities and the (4) systematic approach to components, systems and networks integration and interoperability. Within smart cities, these tasks are far more challenging given the plethora of technologies including legacy technology, data models, communication protocols, components, modules, providers, vendors, use cases, stakeholders, application domains, services and users, which are involved and should be considered.

The overall structure of the paper and its contributions is provided in Figure 1 with the concept inputs on the left and derived results for the smart city community and interested stakeholders on the right. The approach is to take various aspects from current theoretical smart city frameworks as an input, including blueprints for ICT reference architectures and belonging standards, various views and dimensions of the sustainability topic, theoretical foundations from the domain of dependable ICT systems, state-of-the-art and emerging technologies as well as theoretical artifacts from the area of quality assurance and testing for communication-based systems. All these inputs are combined within the paper as to provide an overall approach for a resilient and sustainable ICT-based smart city. This resilient and sustainable smart city should build on three main pillars in our view, which are depicted as the papers' theoretical output on the right side in Figure 1, namely: the design principles for developing sustainable and resilient ICT for smart cities, concrete recommendations for next steps on technological and organizational level, as well as the concept for continuous quality assurance and certification processes for the establishment of high quality and secure critical IT and data communication infrastructures within an urban environment.

A key goal of this paper is to combine the reasonable features from multiple approaches to ICT reference architectures and to show how these can be used to define a sustainable and resilient ICT infrastructure within a city/community. Such a reference architecture for ICT in smart cities emerged from the activities of established standardization groups such as the DIN 91357 [2] by the German Institute for Standardization,

which is based on the European Innovation Partnership (EIP) and the Memorandum of Understanding (MoU) on Sustainable Cities and Communities (SCC) [3] and its belonging ICT reference architecture work stream [4]. In this context, we also discuss the quality assurance approach oupPLUS and detail the way it can be used to provide an interoperable, secure and resilient ICT. oupPLUS is a quality-oriented extension of open urban platforms as defined in [2] and was initially presented in [5,6]). Furthermore, we outline key technological pillars which should be considered in each smart city development plan and for which oupPLUS provides the means for quality assurance, resilience and sustainability.

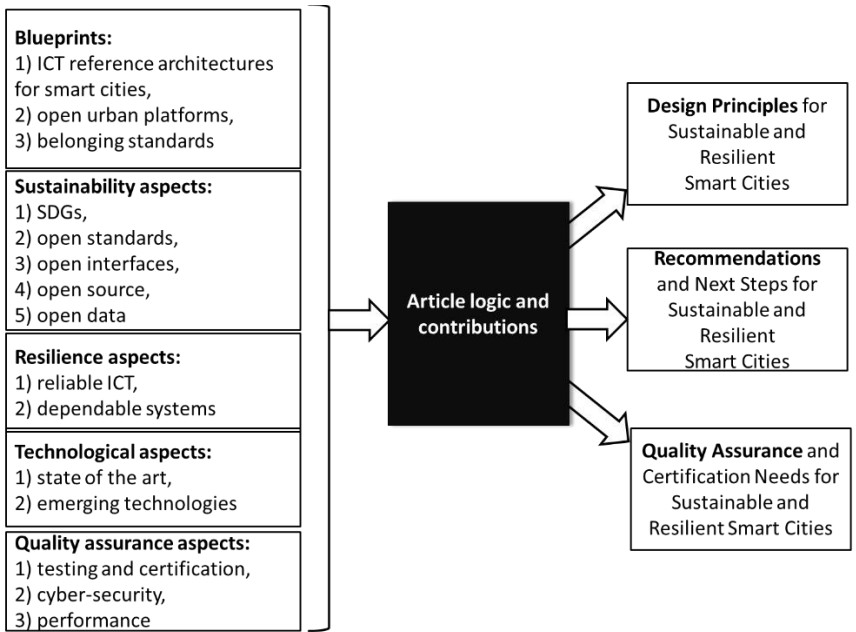

**Figure 1.** The contributions of this paper—(1) inputs on the (**left**) and (2) outputs on the (**right**).

The rest of this paper is organized as follows: Section 2 reviews various related research and development activities including long-term initiatives such as FI-WARE (FI-WARE stands for Future Internet-ware) or European projects such as Triangulum and Espresso. Section 3 provides the basic definitions of sustainability, smart cities and sustainable smart cities, thereby setting a frame around the discussions and elucidations of the current paper. Section 4 presents the concept of reliable ICT and motivates it through some historical definitions. Section 5 sets the ICT frame for further discussions by presenting the concept of ICT reference architectures and open urban platforms, which is the natural setting for the development of a smart city and enhancing its reliability. Section 6 shows the importance of quality assurance within urban ICT and proposes a special ICT reference architecture that provides the necessary constructs for systematically achieving a high degree of quality assessment and certification for smart city technology. Section 7 presents key technological developments of relevance for future smart cities and places them in the framework of an Open Urban Platform. The final two sections—Sections 8 and 9—depict the recommendations and conclusions for achieving reliable and sustainable smart cities, thereby concluding and wrapping up the paper.

## 2. Related Work

This section presents related work relevant for the conceptual and technical understanding of ICT in smart cities.

### 2.1. General Smart City Theory

A large number of smart city solutions have been developed across European and German cities in recent years. Ref. [7] provides a systematic view to structuring and

understanding the overall smart city picture including innovation eco-systems, societal challenges and looking beyond the technological aspects, which are actually to a large extent at the heart of the current paper. A similar notable piece of research exploring the scientific big picture of smart cities is provided in [8,9]. In addition, when approaching and understanding the overall urban landscape and innovation eco-systems, various meta-studies summarizing and comparing scientific and hands-on experiences are of great help, with [10] being such examples worth mentioning.

Refs. [11–13] are some examples of our previous work from the domains of mobility and energy, which enable the sharing of mobility resources (e.g., car and bike sharing) as well as the optimized utilization of energy sources subject to certain economic and eco-logical goals. Furthermore, various research activities were conducted towards analyzing the emerging relationships within future smart cities and the belonging stakeholders. For example, [14] analyzes the role of universities in such an envisioned urban eco-system engagement approach.

Beyond the above research activities, we can observe that the digitalization and optimization of urban processes is at stake, since it has reached limits when it comes to single institutions providing smart services in a city. Hence, it is obvious that a horizontal approach is required that will break the various silos and facilitate the collaboration between different stakeholders, players and citizens within an urban environment. The driver behind such a horizontal approach is clearly the ICT, which should aim at exposing various types of data and enabling the exchange of information between a large number of currently involved and potential players in an urban environment. Hence, it should not only optimize existing courses of action, but it should also provide a platform for the creation of future processes and cases of use aiming at improving quality of life in future smart cities.

### 2.2. Knowledge Management for Smart Cities

An important related research direction to be considered is knowledge management for smart cities and urban innovation [15]. The idea in this area is to collect and share knowledge regarding different projects, initiatives and concepts on organizational (e.g., with regard to human resources management [16]) and technical level (e.g., IoT activi-ties 17), in order to arrive at a collaborative approach towards the development of future urban environments. Thus, cities are expected to become knowledge hubs for different participants in an urban innovation eco-system [17,18]. The involved stakeholders who are considered for collaboration involve the public and the private sector. This includes city representatives and departments, citizens, (utility) companies, universities and research institutes as well as large scale industry such as software power houses or telecom and infrastructure providers.

### 2.3. Reference Models

In order to achieve such a horizontal approach, a number of reference models have been proposed within the past years, e.g., EIP SCC [4], DIN OUP [2,19–21], the Espresso reference architecture [22,23], the reference architecture from the STREETLIFE project [24], the IoT-based reference model of [25], the corresponding model from the Triangulum project [26–28] and more. These models, in general, structure the city in layers that reflect the various levels of information processing, starting from the data sources, continuing with the communication network, the cloud/edge databases and the belonging data analytics up to the level of applications and services, which emerge based on these data processing chains. Many of these reference architectures come up with design principles such as open interfaces, open standards, open source, open data and artifacts that aim at avoiding vendor lock-in and creating viable local eco-systems for ICT developments and innovation.

### 2.4. ICT Smart City Platforms and Solutions

Particular solutions implementing aspects of the above reference models are given by FI-WARE [29], UrbanPulse [30], DKSR [31], CKAN/DKAN [32,33], MindSphere [34] and many others, which are currently being systematically examined in a study supported by the Morgenstadt smart city community in Germany [35]. Thereby, over 60 different commercial and open source urban platforms are analyzed in terms of their compliance to the key features of the above mentioned reference architectures and with regard to their openness and capability to enable smart city eco-systems (including startups, small and medium-sized companies, open source initiatives, industry, and academia).

### 2.5. Data Models and Communication Protocols

In addition to the aspects of reference architectures, a large number of data models and communication protocols are of particular relevance for the ICT of a smart city. First of all, the standard protocol suites of the Internet are of particular relevance—these suites include protocols such as Internet Protocol Version 4/Version 6 (IPv4/v6), Open Shortest Path First (OSPF), Routing Information Protocol next generation (RIP-ng), Border Gateway Protocol (BGP), Differentiated Services (DiffServ), Integrated Services (IntServ), Address Resolution Protocol (ARP), Neighbor Discovery Protocol (ND), Dynamic Host Configuration Protocol (DHCP), Domain Network System (DNS), Network Time Protocol (NTP), Hypertext Transfer Protocol (HTTP), Simple Object Access Protocol (SOAP) and many others, which ensure fundamental network functions of forwarding, routing, QoS (Quality of Service) and name resolution to name some (see, e.g., [36]). In the IoT access network segment, relevant protocols are given by ZigBee [36], low-rate wireless personal area networks (IEEE 802.15.4), IPv6 over Low power Wireless Personal Area Network (6LowPAN) or low power Long Range Wide Area Network (LoRaWAN), whilst the over spanning IoT architectures utilize means such as Constrained Application Protocol (CoAP) or Message Queuing Telemetry Transport (MQTT) (see, e.g., [37]). All these protocols are based on well-defined non-proprietary standards enabling open communication network exchange within a smart city. The communication networks carry data using data and information models such Next Generation Sensor Initiative – Linked Data (NGSI-LD), Sensor Model Language (SensorML), Data Catalog Vocabulary – Application Profile (DCAT-AP) as well as different linked data-based formats describing the data and information within a smart city (see, e.g., [38,39]).

## 3. Sustainable Smart Cities

This section presents fundamental concepts for smart cities and provides working definitions.

### 3.1. Definition: Sustainability

There are a large number of definitions for Sustainability in the literature. Ref. [40] provides, e.g., a summary and overview of definitions from social and environmental perspectives. Many of the definitions are not straight forward but aim at discussing and putting things in various perspectives as to encompass all relevant aspects for a particular field. In view of ICT for smart cities we use the term in the following sense: *"Sustainability is the capability of a system to exist in the long term based on its modularity, flexibility and intensive interaction and balanced exchange with its eco-system/environment".*

Yet, the overall term of Sustainability has been structured along the Sustainable Development Goals (SDGs), which are illustrated in Figure 2. The SDGs were defined by the United Nations in 2015. Significant progress in this regard needs to be achieved by 2030 [41].

Herein, SDG 11 is directly framed to be Sustainable Smart Cities and Communities. In addition, ICT for smart cities can directly contribute to SDG 9—Industry, Innovation and Infrastructure, SDG 8—Decent Work and Economic Growth, SDG 12—Responsible Consumption and Production, and SDG 7—Affordable and Clean Energy, and indirectly to SDG 1—No Poverty, SDG 2—Zero Hunger, SDG 3—Good Health and Well-Being, SDG

6—Clean Water and Sanitation, SDG 13—Climate Action and SDG-16—Peace, Justice and Strong Institutions. An increased utilization of ICT within smart cities can positively influence a number of key areas such as mobility, (renewable) energy, health, transportation, water-management, waste-management, circular processes, public administration, or public safety and thereby contributing in parallel to job creation, economic growth, increased quality of life and poverty reduction. Hence, we believe that ICT-based smart cities are to be seen as a means for achieving the above listed SDGs [42]. Thereby, ICT tools and automations introduce additional intelligence in the city processes, which can be efficiently utilized towards achieving the SDGs and improving quality of life in cities, communities and across the globe.

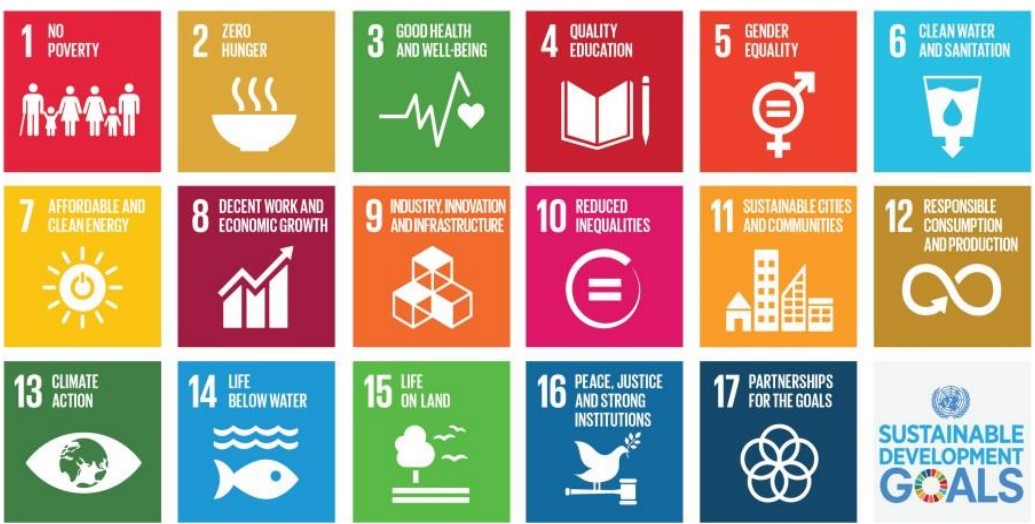

**Figure 2.** The Sustainable Development Goals (SDGs) according to [41].

### 3.2. Definition: Smart Cities

We summarize the smart city understanding according to [19–21] as follows. A smart city is:

1. Intelligent
2. Sustainable, but also
3. Adaptable meaning that it can adapt its action and process options according to social and/or economic needs,
4. User-oriented, meaning that the citizens of a municipality are at the center of attention; satisfying their needs and optimizing related processes and services using ICT is the main goal of a smart city
5. Responsive, meaning that both the administration and the optimized processes and ICT services are in constant interaction with the citizens of a smart municipality
6. Sensitive, using various types of sensor technology and data acquisition tools, a constant attempt is made to scan the situation and obtain the relevant data and to use it for new types of services, applications and process control options, and
7. Innovative, meaning that the smart city creates an eco-system in which constant innovation—based on data, information, networking and modern ICT—continuously optimizes and improves urban efficiency and the quality of life of citizens.

### 3.3. Definition: Sustainable Smart Cities

In conclusion of the above, a smart city needs to be intrinsically sustainable. We perceive a sustainable smart city to be one that aims at fulfilling the corresponding SDGs and tries to achieve this in a sustainable way, and based on a sustainable infrastructure (e.g., Internet and telecom networks, mobile networks, Wireless Fidelity (WiFi), IoT, urban

data spaces etc.). This means that the processes (e.g., public transportation processes like ticketing, parking spot management, on-demand waste management, demand-response flexibility exchange in energy cooperatives, inter- and multimodal mobility etc.) and the infrastructure of a city/community should work in a way so as to achieve the SDG goals (as discussed in Section 3.1). It should be designed to be modular, flexible and also be able to successfully deliver results in the long-term, in correspondence to the latest technological developments as well as social and environmental requirements.

Based on the above definitions and elucidations, we can see that the term smart city has a very broad scope. However, provided that ICT including IT are considered as the efficient boost towards the development of future smart cities in the various areas like health, buildings, citizens' participation, mobility, transportation etc., the following chapters focus strongly on the IT, telecommunication and data foundations of an urban infrastructure as a prerequisite for resilient and sustainable smart cities.

## 4. Reliable ICT

In this section, we relate the requirements of the urban ICT of a sustainable smart city to the methods and means to make urban ICT reliable, dependable, and as a result trustworthy, such that it can constitute the infrastructural basis of a sustainable smart city.

### 4.1. Definition: Reliability

Similarly to the term sustainability, the notion of reliability has been the subject of many different definitions which are overlapping to a large extent and mostly reflect similar aspects in different domains. Furthermore, reliability can also be seen as an attribute of dependability as we are going to briefly discuss in the next subsection.

According to the Cambridge online dictionary, the term reliability is defined as follows: *"The quality of being able to be trusted or believed because of working or behaving well"* [43]. With respect to information and communication systems, which are at the heart of a smart city, the following definition is preferred: *"Reliability is defined as the probability that a product, system, or service will perform its intended function adequately for a specified period of time, or will operate in a defined environment without failure"* [44]. This definition originates from the American Society for Quality (ASQ) and has a more technical perspective. Hence, by applying this definition to the ICT infrastructure of a smart city, we put the reliability requirements in context.

### 4.2. Definition: Dependability

The term reliability can also be seen as an attribute of the overarching concept of dependability, which is important in the area of distributed systems and communication networks. Dependability and dependable systems have been discussed in many fundamental research studies [45–48]. Thereby, dependability encompasses the various types of non-functional requirements for designing high-quality systems as well as the means to achieving these requirements and the major threats or impairments in the ICT context. Ref. [45] provides a comprehensive illustration of the key components comprising dependability (see Figure 3). This includes, among others, key security aspects such as availability, integrity and confidentiality. They play a vital role for urban ICT since only trustworthy processes and infrastructures will be adopted widely.

### 4.3. Quality Assurance for ICT Technology

In view of trustworthiness as well as of effectiveness and efficiency, quality assurance for urban ICT is essential for achieving reliable ICT components for smart cities. There has been intensive research on this topic in recent decades. Along this, the industry has established processes for quality assurance of single components, systems and of the overall systemic approach. Essentially, quality assurance addresses two main aspects: (1) the processes when developing/creating complex systems and (2) the testing of complex systems for deployment and operation, which ideally should take place in parallel to the develop-

ment processes. Testing comprises various validation and verification activities for ICT and constitutes the most important quality assurance method in industry [49]. In testing, it is important to address different functional and non-functional perspectives on single components as well as on complex urban ICT systems as a whole. Testing approaches include conformance testing (i.e., the conformance to standards and requirements), interoperability testing, load and performance testing, security and penetration testing as well as usability and acceptance testing [50]. All these facets need to be delivered by urban ICT infrastructures on a reasonably high level in order to guarantee the success of the intended introduction of smart city development principles. These aspects are central for the reliability and resilience of the ICT infrastructure acting as a smart city backbone.

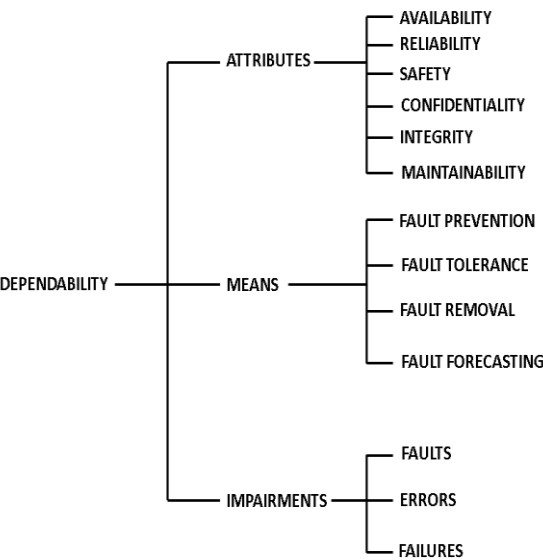

**Figure 3.** The structure of the dependability concept according to [45].

## 5. ICT Reference Architectures and Open Urban Platform for Smart Cities

Reference architectures as discussed in Section 2.3 are seen as propositions for blueprints of the ICT infrastructure in smart cities. ICT reference architectures have been developed along the concept of Open Urban Platforms (OUP), which have initially been worked out on a European level [3,4] and subsequently standardized [2] by standardization bodies such as the DIN in Germany. OUP is a special type of blueprint that postulates a very high level of openness for urban ICT by prescribing the utilization of open interfaces and open standards for the data and information exchange between infrastructure components, systems and processes in a smart sity. Furthermore, OUP advocates the usage of open data and open source as key data sharing and code transparency principles. However, these two principles are mere recommendations, as opposed to the open interfaces/standards requirement that is the key prerequisite for establishing an ICT infrastructure in conformance to OUP.

Further tasks of an urban ICT reference architecture include the provisioning of a unified view on the ICT strategy of the city in question, as well as by the elucidation of the interfaces between the various technological layers and components within a city. Another important aspect is the need to be able to accommodate existing systems under the overall umbrella of an ICT reference architecture. Especially within the scope of open urban platforms, one of the main goals is the intention to enable the replication of smart city solutions across cities, thereby creating an eco-system and a new market in order to boost smart city approaches in terms of functionalities, deployments and applications.

Figure 4 provides a high-level view on the standardized Open Urban Platform concept. The overall structure is constituted by a set of layers describing the information processing paths within a city, roughly starting from the data sources (e.g., sensors), continuing with

the communication network that enables the overall city interconnectivity, the databases and cloud infrastructures in the backend and the services which utilize the urban data in order to provide added value to various stakeholders such as the city/community, companies and the citizens. The overall layered structure is accompanied by cross-layer aspects relating to privacy and security as well to the management of the overall infrastructure and its common services.

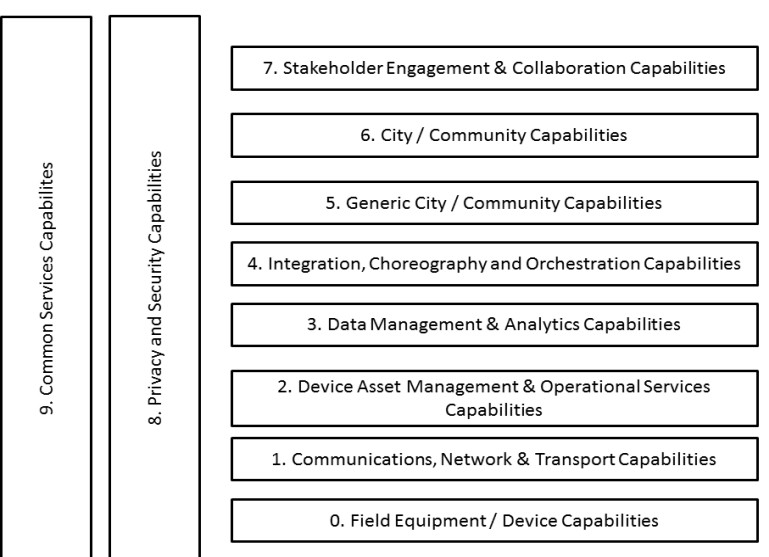

**Figure 4.** The high-level view on the open urban platform concept as defined in [2,4].

The detailed OUP definition in [4] provides various so-called capabilities for each layer/pillar that basically capture the functionality and the aspects which are to be implemented in this layer/pillar. Such capabilities are for example (Complex) Event Processing and Metadata Management for the "3. Data Management and Analytics Capabilities" layer or by Sensoring and Measuring and Time and Position Keeping for the "0. Field Equipment/Device Capabilities" layer.

## 6. oupPLUS: The Quality Assurance View on Smart Cities

The OUP extension oupPLUS (see [5,6]) has been developed by Fraunhofer FOKUS in order to provide quality assurance means enabling security, trustworthiness and reliability for ICT in smart cities. It brings together the insights from the Triangulum reference model [26–28] and the Espresso reference architecture definitions [22,23]. Based on a strong background in quality engineering for ICT systems, oupPLUS enables a systematic quality process towards establishing a reliable ICT backbone for a smart city.

The overview of oupPLUS is provided in Figure 5. oupPLUS resembles many of the OUP layers in addition to emphasizing on the key principles of open interfaces and open standards. Beyond this, oupPLUS aspires towards the identification and specification of abstract open interfaces as extensions to the reviewed and unified architectural principles of the considered research and standardization results. Thereby, oupPLUS aims to establish guidelines for well-defined open interfaces between the different ICT components and layers of an ICT smart city backbone. This opens a variety of new opportunities for the smart city development across the globe, such as (1) interoperability of different solutions in various areas and urban environments, (2) replication and reuse of smart city solutions across multiple cities, (3) the creation of a viable ICT eco-system in cities/communities including the participation of small and medium-sized enterprises, (4) the increased utilization of open source, (5) avoiding vendor lock-in and last but not least (6) the systematic quality assurance of ICT components, solutions and complex systems within/for a smart city.

**Table 1.** SAPs and examples for their implementation by protocols.

| SAP-Name: Description | Possible Protocols (Protocol Stacks) and Additional Standards for the SAP Implementation |
|---|---|
| CTL-SAP: Control SAP | Cloud control protocols: (1) OpenStack Representational State Transfer (REST) Application Programming Interface (API), (2) SOAP (HTTP over TCP/IP or User Datagram Protocol (UDP)), (3) Common Object Request Broker Architecture (CORBA) with the Internet Inter-ORB protocol (IIOP) over TCP/IP, (4) Transport Layer Security (TLS) over Worldwide Interoperability for Microwave Access (WiMAX), Universal Mobile Telecommunications System (UMTS), Long Term Evolution (LTE), or Wireless Fidelity (WiFi) <br> Database/Data warehouse Control Protocol (Stack): (1) Structured Query Language (SQL) commands, (2) Open Network Computing (ONC) Remote Procedure Call (RPC), (3) Open Cloud Computing Interface for Cloud API <br> Further Standards: (1) Open Charge Point Protocol (OCPP) for communication with Charging Management System (CMS) and charging station, (2) Open Smart Charging Protocol 1.0 (for 24h-prediction integration) for smart grid and CMS, (3) Smart Energy Profile Application Protocol (P2030.5) |
| MTD-SAP: Metadata SAP | Metadata harvesting protocols: (1) Open Archives Initiative Protocol for Metadata Harvesting (OAI-PMH) using the Extensible Markup Language (XML) (2) Web Catalogue Service (CSW) of the Infrastructure for Spatial Information in the European Community (INSPIRE) <br> Service and metadata discovery protocols: (1) Message Protocol v4.0 of the centrally managed distributed data exchange layer (X-ROAD), (2) Object Name Service (ONS) for authoritative metadata and services associated with a given id-key <br> API for meta-data access: (1) CKAN API using JavaScript Object Notation (JSON) <br> Meta-models: (1) Data Catalog Vocabulary (DCAT), (2) Common Warehouse Meta-model (CWM), (3) DCAT Application Profile (DCAT-AP), (4) Open Graph Protocol (OGP) for describing Web objects, (5) Resource Description Framework (RDF) |
| DX-SAP-Data: Exchange SAP | Data Transfers Protocol Stack: (1) Remote Direct Memory Access protocol over Converged Ethernet (RoCE), (2) FTP or SFTP or others/WiMAX directed wireless using orthogonal frequency division multiplexing (OFDM), (3) NFS (Network File System), (4) Kafka (data and meta-data) API [51] |
| DSD-SAP: Data Sources Data SAP | Standard Messaging Protocol Stack: (1) direct device-to-device messaging with Constrained Application Protocol (CoAP) optimized for IoT, (2) Message Queuing Telemetry Transport (MQTT), (3) Advanced Message Queuing Protocol (AMQP) for peer-to-peer (P2P) and publish/subscribe communication, (4) Thread [52] over TCP/IP, (5) telecontrol (IEC 60870-5-104), (6) Data Distribution Service (DDS) for real-time systems, (7) Next Generation Sensor Initiative – Linked Data (NGSI-LD) <br> Wireless Messaging Protocols: (1) CoAP, MQTT, AMQP, threads or IEC 60870-5-104 over standard mobile networks such as General Packet Radio Service (GPRS), WiMAX, LTE or UMTS <br> Streaming Data Protocols: (1) Advanced Video Coding (AVS) with H.246 over Real-Time Transport Protocol (RTP) for video/streaming data <br> Stream Establishment/Connection Negotiation/Control: (1) Session Initiation Protocol (SIP), Session Description Protocol (SDP), (2) (Secure) RealTime Control Protocol ((S)RTCP) for secure connection negotiation), (3) Real-Time Streaming Protocol (RTSP) for video play pause control, (4) Program and System Information Protocol (PSIP) for television (and other) metadata |
| RWD-SAP: Raw Data SAP | Short Range Wireless Small Device Messaging Protocols: (1) IoT protocols for small devices, e.g., sensor including CoAP, Routing Protocol for Low-Power and Lossy Networks (RPL), or IPv6 over Low power Wireless Personal Area Network (6LoWPAN), (2) ZigBee [37], (3) MQTT for Sensor networks (MQTT-SN), (4) MQTT as legacy solution with bigger overhead, 5) Thread or Z-wave [53] to connect and control products at home <br> Wide Range Wireless Small Device Messaging Protocol: (1) LoRaWAN for wireless battery-operated things in regional, national or global network, (2) narrowband IoT (NB-IoT), (3) LTE Machine Type Communication (LTE-MTC), (4) Extended Coverage GSM IoT (EC-GSM-IoT) <br> Wired Small Device Messaging Protocols: (1) Modbus [54], (2) Controller Area Network (CAN), (3) RS-485 protocol by the Telecommunications Industry Association and Electronic Industries Alliance (TIA/EIA) <br> Specific Raw Data Protocols: (1) low energy Bluetooth for control of video signal, (2) Lightweight machine-to-machine communication (LwM2M) of the Open Mobile Alliance (OMA) for coordination of small devices, (3) Simple Sensor Interface (SSI) for simple direct access from PCs to sensors, (4) Lightweight Local Automation Protocol (LLAP) <br> Further Standards: (1) Direct Charging (DC) charging points (IEC 61851-24), (2) Sensor Model Language (SensorML), (3) NGSI-LD |
| MGM-SAP: Machine Management SAP | Static Device Management Protocols (Stack): (1) Secure Shell over TCP or UDP (widely used to have a direct connection with a device/PC/server/machine), (2) Telnet for interactive text-oriented communication (legacy solution, pre-SSH), (3) TR-069 for remote management of customer-premises equipment by the Broadband Forum, (4) Simple Network Management Protocol (SNMP), (5) Network Configuration Protocol (NETCONF) <br> Mobile/Small Device Management Protocols (Stack): (1) Mobile Device Management (MDM) protocol for dedicated mobile device management [55], (2) OMA Device Management via XML with the Wireless Session Protocol (WSP) or Wireless Application Protocol (WAP) over HTTP, the Object Exchange (OBEX) protocol and over wireline like Universal Serial Bus (USB) or Serial Interface (RS-232) and/or over wireless media like Global System for Mobile Communications (GSM), Code-division multiple access (CDMA), Infrared Data Association (IrDA), or Bluetooth), (3) LwM2M, (4) Java 2 enterprise edition (J2EE) Mobile Device Management and Monitoring (JSR 233), (5) Open Trust Protocol (OTrP) <br> Network Management Protocols Stack: (1) SNMP, (2) TR-069 |
| BCN-SAP: Beacon and Near Field SAP | Bilateral/Cluster Beacon and Near Field protocols: (1) Hypercat [56], (2) Physical Web, (3) Multicast Domain Name System (mDNS), (4) Universal Plug and Play (UPnP) <br> Unilateral Beacon and Near Field protocols: (1) Near Field Communication (NFC) (newer, more data can be transferred), (2) Radio-Frequency Identification (RFID) (widely used) <br> Other Bilateral/Cluster BCN Protocols: (1) Time Synchronized Mesh Protocol (TSMP), (2) LwM2M v1.0 for coordination |

**Table 1.** *Cont.*

| SAP-Name: Description | Possible Protocols (Protocol Stacks) and Additional Standards for the SAP Implementation |
|---|---|
| RtCD-SAP: Routing Coordination and Discovery SAP | Network Layer Protocols: (1) Internet Control Message Protocol (ICMP) for IPv4/v6, (2) IPv4/v6, (3) Internet Protocol Security (IPSec), (4) Multiprotocol Label Switching (MPLS) <br> Routing Protocols: (1) Routing Information Protocol (RIP) or RIP next generation (RIP-ng), (2) Open Shortest Path First (OSPF v3), (3) Border Gateway Protocol (BGP v4), (4) Intermediate System to Intermediate System routing protocol (IS-IS) <br> Other network coordination protocols: (1) Network Time Protocol, (2) DHCP(v6), (3) Neighbor Discovery Protocol (ND), (4) Address Resolution Protocol (ARP), (5) Domain Name System (DNS) |
| ATH-SAP: Authentication/Authorization SAP | Authentication/Authorization Protocols: (1) Kerberos [57], (2) Remote Authentication Dial-In User Service (Radius), (3) Diameter for authentication, authorization, and accounting protocol in computer networks (a pre-Radius authentication), (4) Public Key Infrastructure (PKI) Authentication, (5) Lightweight Directory Access Protocol (LDAP), (6) eXtensible Access Control Markup Language (XACML), (7) Access Control List (ACL), (8) OpenID for open standard and decentralized authentication, (9) OAuth for open standard and decentralized authorization |

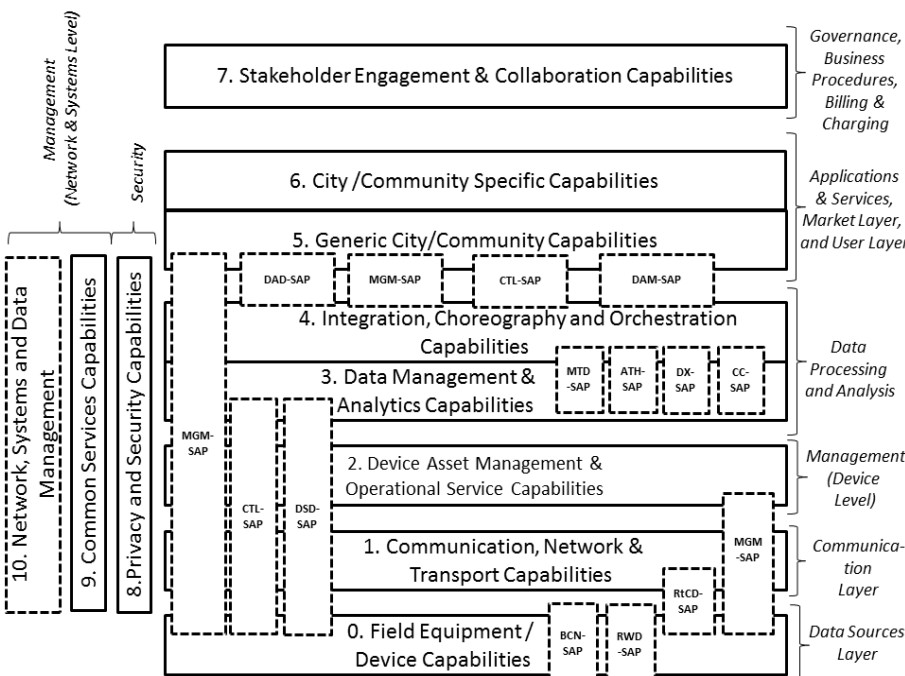

**Figure 5.** The oupPLUS architecture providing a quality-oriented extension of open urban platforms as defined in [6]. SAP stands for Service Access Point. Most of the SAPs used in this figure are described in Table 1. The other SAPs are the Data Analytics Data SAP (DAD), Data Analytics Management SAP (DAM), and the Cluster/Cloud Control SAP (CC).

The above-mentioned abstract open interfaces are constituted by the Service Access Points (SAPs) in Figure 5. The term SAP is borrowed from the Internet/telecommunication domain (especially TCP/IP and ISO/OSI). The SAPs of oupPLUS are abstract interconnections between various layers with their belonging capabilities and instantiating components in a real urban ICT instantiation of oupPLUS. Thereby, the SAPs normally connect between modules, components, or services. A particular instantiation of an SAP is given by a whole communication stack instance including a physical-, link-, network- and service-layer protocol, such as Ethernet/IPv6/TCP/HTTP/DCAT-AP to give an example. More detailed descriptions on the single SAPs are provided in [6]. However, in Table 1 we provide a brief introduction of the SAPs from Figure 5 and list relevant protocols stacks and standards which can play a role within a smart city ICT backbone. Furthermore, the information in Table 1 already hints on how oupPLUS can be used as a concept for Quality Assurance, namely by simply "testing the SAPs" which automatically means the testing of the relevant protocol stacks. In this setup, all possible types of test examinations—conformance-, interoperability-, security/penetration-, load- and performance-testing—can be executed

leading to the systematic approach to high quality ICT components and complex systems. The aspect of Quality Assurance is further detailed in Section 7.2.

## 7. Technologies of Relevance for oupPLUS

This section provides a deep dive into the technologies being relevant for oupPLUS.

### 7.1. Overview of Smart City Information and Communication Technologies

Main pillars for future urban ICT eco-systems include: (1) IoT sensors and IoT communication protocols, (2) the 5th and 6th generation of mobile communication (5G/6G), (3) public WiFi, (4) fiber infrastructure, (5) cloud/edge infrastructure, (6) open and/or big data, (7) geographic information systems, (8) data analytics and artificial intelligence (AI), (9) city dashboards, (10) end user applications and services and (11) data governance and sovereignty. All of these can be mapped to the layers of the oupPLUS reference model described in Section 6. Together they can form different instances of ICT solutions and/or platforms in an urban infrastructure. Thereby, all of these technologies can be combined and configured to conduct different tasks within a smart city reference model. The technologies listed in Table 2 offer open interfaces and standards, which make it possible to map them to the capabilities of OUP/oupPLUS and embed them into an urban ICT eco-system via the corresponding open interfaces. These technologies are subject to further development in the ICT research domain.

### 7.2. Quality Assurance Technologies and Principles

As discussed in Section 4, systematic quality assurance is the key to achieving trustworthy and resilient integrated smart city infrastructures. This is best achieved by first providing an overview of various available urban ICT solutions and components and secondly providing the means for systematically testing and certifying those solutions/components.

oupPLUS provides a meta-model for the ICT infrastructure in a city, in order to structure the quality assurance for an urban ICT (see Figure 6). It can be used to describe different potential components and solutions by assigning them to different layers/pillar as well as belonging SAPs and concrete interface/protocol stacks as described in Section 6. Furthermore, a component/solution/product should only be accepted (and deployed) when it adheres to the principles defined in the relevant OUP standards, such as the utilization of open interfaces based on open standards. Figure 6 contains an excerpt of an oupPLUS ontology for the purpose of describing smart city solutions, components and overall ICT eco-systems based on principles of oupPLUS. In Figure 6, we see all artifacts from Figure 5 including the OUP layers, the capability maps for a solution/component, the SAPs from oupPLUS as well as the abstract interface/protocol stacks for communication over an SAP, as described in Section 6 and exemplified with concrete instances in Table 1. Hence, Figure 6 provides the basic artifacts which are required in order to start cataloguing and describing urban ICT infrastructures in addition to the illustrations in Figure 7. These illustrations provide on one hand a more elaborated description of the technical aspects (SAPs, interface/protocol stacks, etc.) whilst at the same time showing the relation to a business perspective including the notion of product and further business views when designing an urban ICT. These extensions and integrational aspects were developed in collaboration with a German smart city portal for creating a supply/demand business community for smart city solutions. Therefore, the oupPLUS meta-model descriptions support the conceptual integration or orchestration of organizational, governmental or business perspectives, which is typically required for a smart city.

In order to achieve a complete approach towards the systematic quality assurance and testing of urban ICT, one last ingredient is missing. The oupPLUS-based description of an ICT infrastructure typically contains many solutions/components/products mapped to layers/pillars, SAPs and interfaces/protocol stacks. Therefore, systematic tests at the SAPs with regard to the required interfaces and protocol stacks are needed. By that, the conformance to standards and requirements, interoperability, security, performance and resilience

of the components and systems as well as the trustworthiness of the overall solutions and the particular urban ICT infrastructure as a whole are assured. Table 3 provides examples of available automated test suites, which can be used at the oupPLUS SAPs. The testing approaches are to be accompanied by practical and appropriate certification approaches for urban ICT. ICT solutions for smart cities should be deployed only if the required quality has been validated, in order to guarantee an overall level of resilience and trustworthiness of a smart city infrastructure.

**Table 2.** Relevant technologies for smart cities.

| Technology | Description |
|---|---|
| IoT sensors and communication protocols | Devices that can measure aspects such as temperature, humidity, gas levels, and infrared radiation (but are not limited to these) and can communicate measured values to an IoT platform are termed as IoT sensors. For the IoT sensors and actuators to function in tandem with the belonging IoT platform (in the backend/cloud), it is necessary that they are able to establish communication channels to the platform, and understand each other's transmitted messages. This is possible when the communication between the various components follows a prior agreed format for self-identification, peer discovery, device management and data transfer among other aspects. Typical protocols from this domain are given by CoAP, MQTT, LoRaWAN, ZigBee, IEEE 802.15.4, NB-IoT, Sigfox [58] and 6LowPAN, to name some examples. |
| 5G/6G | 5G is the fifth generation of standards for cellular wireless communication. With the 5G infrastructure in place, the data transfer rates will multiply by about 100 times offering network latencies as low as 1–10 ms and 1000 times more capacity, while reducing the mobile data delivery costs by a factor of 10 in comparison to the current 4G networks [59–61]. A 5G infrastructure would also support a significantly larger number of concurrent connections, which implies that the same network has the potential to facilitate IoT infrastructures at a large scale suitable for smart cities. The development of 6G, the sixth generation of mobile communication has been started and will bring new features also for smart city solutions. |
| Public WiFi | Public WiFi is a service offer, in which the cities offer free Internet access to their citizens and tourists usually at the most important and popular spots in the city. This indirectly helps retain the WiFi users for a longer period of time, which in turn helps the local businesses. Moreover, it also offers a platform for the city to share various tourist information and promote local events and businesses. |
| Fiber Infrastructure | Optical fibers are the fastest medium of data transmission that offer the highest efficiency and bandwidth. They also support communication over very long distances [62]. In addition to the benefits mentioned so far, the optical fibers are also resistant to electromagnetic interference; so more secured and reliable. They are also much lighter, thinner, flexible and corrosion resistant in comparison to copper wires. |
| Cloud/Edge | Cloud computing is a paradigm that makes automatic on-demand provisioning of various computing resources such as processing power and storage, without the active direct intervention from a user. With cloud computing the resources are physically placed and maintained within large data centers, while the computing resources are shared as per demand among various users over the Internet. Advantages of cloud computing are manifold; some of them are automatic resource provisioning, easy accessibility of resources, virtually unlimited availability of computing power and storage.<br>Edge computing on the other hand is a distributed computing paradigm where the computation is done at or pushed closer to the data source rather than moving the data to a remote centralized processing unit. Edge computing is highly relevant in the context of IoT and 5G. The benefits of this approach are bandwidth savings, improved response times for orchestration and feedback-based systems. |
| Open and/or big data | According to the European Open Data Portal, "Open data are data that anyone can access, use and share. Governments, businesses and individuals can use open data to bring about social, economic and environmental benefits" [63]. It also states that Open Data must be licensed and must allow people to transform, combine, share them for any purpose they deem without any binding restrictions, both commercially and non-commercially. The major sources for open data are, but not limited to, scientific communities, governments and non-profit organizations. On the site of disadvantages, open data might be biased, violate privacy unintentionally, misinterpreted and misused, lead to decisions because of the poor data quality and cause unclear accountability among other possibilities [64].<br>Oxford defines big data as "extremely large data sets that may be analyzed computationally to reveal patterns, trends, and associations, especially relating to human behavior and interactions." Big data is the basis for manifold smart analytics, services and applications in smart cities. Technologically speaking big data also refers to the processes, technical frameworks and tools involved in the data collection and information analysis from the captured data. |
| Geographic Information Systems | Geographic Information Systems (GIS) constitute a technology which has been already in use by public administrations for the past decades, in order to manage different types of information with geographic relevance. GIS systems capture different map structures and manage geographically various objects, for instance properties, sizes of land area etc. There are different relevant formats which capture data and metadata for such GIS systems, with the Infrastructure for Spatial Information in the European Community (INSPIRE) being the most prominent standard set in this area. Examples of successful usages are given by land area management, asset management and tracking, as well as various types of visualizations (e.g., map-based COVID-19 dynamics illustrations). |
| Data Analytics and Artificial Intelligence | Data analytics as the name suggests is the process of cleansing, transforming, examining and visualization of datasets, usually to gather insights that assist in decision making and establish a correlation between the various factors involved. Very often, the integration of data from various sources is a pre-step to data analytics.<br>Artificial intelligence (AI) is a field of computer science, in which machines act upon inputs from their environment by simulating human intelligence, while incrementally learning from the interpretation of the input values. AI can be trained to perform a variety of tasks. Some applications of AI in the scope of IoT are to predict maintenance, automation failures, connectivity issues and intelligent orchestration of tasks in a complex IoT system. |

**Table 2.** *Cont.*

| Technology | Description |
|---|---|
| City Dashboards | City dashboards provide the possibility to gain an overall view on certain aspects of a smart city (area). Typically, city dashboards aggregate data from different sources, including Urban Data Platforms, Open Data portals, GIS systems, IoT platforms and data from commercial data providers (e.g., from mobile network operators). In that sense, city dashboards allow to configure a particular set of canvases that enable the aggregated key performance indicator (KPI) and metrics' monitoring relating to specific areas and aspects such as air quality, traffic congestion, crowd management, energy management, water management and further. |
| End-User Applications and Services | Typical scenarios that use data efficiently include city dashboards and end-user applications and services. These applications and services can either work autonomously on the basis of predictions drawn from models (as for the regulation of traffic and public lightening), give valuable feedback to decision makers as to the success of their policies, or provide incentives for citizens to change their behavior in a manner, which both benefits them and the society. These end-user applications and services can range from very complex and critical systems involved in energy distribution and retention, over business-oriented services as E-Vehicle rental, to simple information applications about the current state of the city, providing information about traffic peak hours, own energy consumption in comparison to other households and suggestions for optimizing the own behavior. |
| Data Governance and Sovereignty | The topic of data governance and data sovereignty are key ingredients in future smart cities. This challenge should be approached on technical and organizational level as well. On organizational level different committees/groups should be setup to regularly review data to be released (e.g., in a municipality) and allow or disallow its publication. On the technical level, emerging concepts such as GAIA-X [65] and the International Data Space (IDS) provide methods and technologies to annotate data sets with belonging usage/utilization rights and to automatically allow or disallow access to data based on these pre-configured rules. Thereby, data are automatically validated before its communication or publishing and belonging policies are evaluated, in order to keep control and guarantee data governance and sovereignty. |

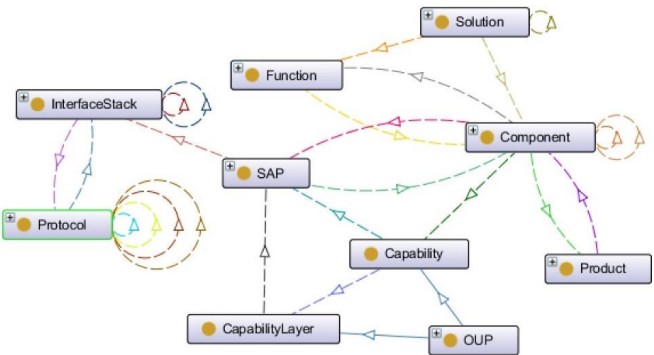

**Figure 6.** The abstract oupPLUS-based ontological structures for describing smart city ICT components and solutions. The used notations originate from the Protégé ontology modelling tool [66].

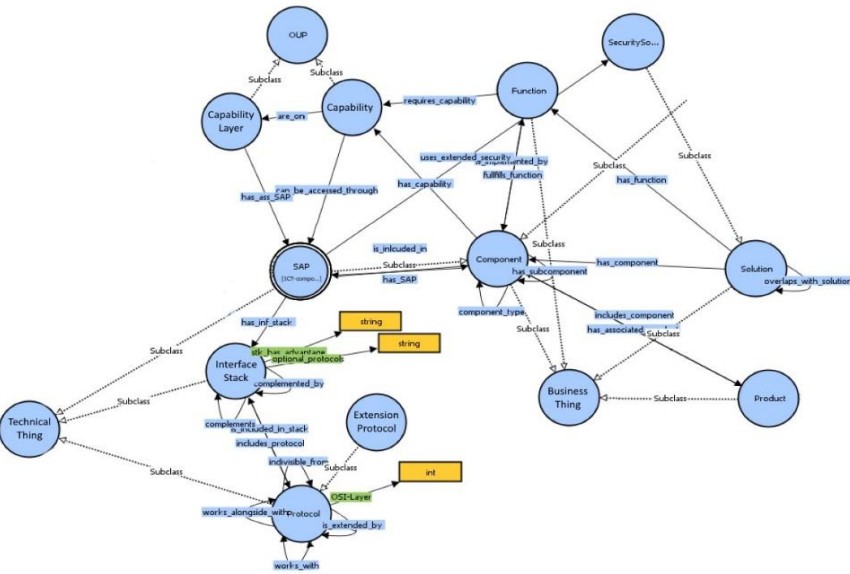

**Figure 7.** Extending the oupPLUS ontology by aspects allowing the integration of business perspectives on smart cities.

**Table 3.** Examples of available test suites that can be applied on the oupPLUS SAPs and further utilized for automated testing and certification.

| Protocol | Test Suite | Type | Platform and Programming Language | Coverage | Provider |
|---|---|---|---|---|---|
| SIP (Session Initiation Protocol) | Conformance Test Specification for SIP | Conformance | TTCN-3 (Testing and Test Control Notation version 3) /Test specification | Protocol conformance: RFC 3261 | ETSI (European Telecommunications Standards Institute), Sophia-Antipolis, France, https://www.etsi.org/ |
| IPv6 (Internet Protocol v6) | IPv6 Ready Logo Program | Conformance/Certification | Test specification and test suite tools | Protocol conformance and interoperability: IPv6 Core Protocols, IPSec (IP Security), IKEv2 (Internet Key Exchange version 2), MIPv6 (Mobile IPv6), NEMO (Network Mobility), DHCPv6 (Dynamic Host Configuration Protocol version 6), SIP (Session Initiation Protocol), IMS UE (IP Multimedia Subsystem User Equipment) Management(SNMP-MIBs), IKEv1, MIPv6 with IKEv1 MLD (Multicast Listener Discovery) | IPv6 Forum, https://www.ipv6forum.com/ |
| IPv6 | IPv6 Test Suites Conformance Test Suites | Conformance | TTCN-3/C++ | Protocol conformance: DHCPv6 (Dynamic Host Configuration Protocol) (RFC3315, RFC3646, RFC3736), RIPng (Routing Information Protocol next generation) (RFC2080) | IRISA (Research Institute Computer And Systems Aléatoires), Rennes, France, http://www.irisa.fr/ |
| IPsec IP Security) | IPv6 Ready Logo Program Phase-2 | Conformance/Certification | Test specification and test suite tools | Protocol conformance and interoperability: IPsec test specification (v1.11.0), IPsec interoperability test scenario (v1.11.0), IPSec test tools (see above), IOL INTACT (v2.0.0b) (Improving Networks Through Automated Conformance Testing, University of New Hampshire, Interoperability Laboratory), self-test tools) | IPv6 Forum, https://www.ipv6forum.com/ |
| SOAP (Simple Object Access Protocol) | SOAP v1.2 Specification Assertions and Test Collection | Conformance | Test specification | Protocol conformance: SOAP 1.2 | W3C (World Wide Web Consortium), Sophia-Antipolis, France, https://www.w3.org/ |
| HTTP (Hypertext Transfer Protocol) | Jigsaw A set of HTTP/1.1 features | Conformance | Test tool | Protocol conformance: HTTP/1.1, Chunk Encoding, Connection Cache-Control, Content-MD5 (message-digest algorithm), Retry-After (delay), Retry-After (date), 300 Multiple Choices, 414 Request-URI (Uniform Resource Identifier) Too Long, Redirect test page, Basic Authentication test, Digest Authentication test, Content-Location test | W3C (World Wide Web Consortium), Sophia-Antipolis, France https://www.w3.org/ |

**Table 3.** *Cont.*

| Protocol | Test Suite | Type | Platform and Programming Language | Coverage | Provider |
|---|---|---|---|---|---|
| MQTT (Message Queuing Telemetry Transport) | IoT Testware: MQTT Test Suite | Conformance | Test specifications and TTCN-3 | Protocol conformance: MQTT v3.1.1 | Eclipse IoT Testware project, https://projects.eclipse.org/proposals/eclipse-iot-testware |
| CoAP (Constrained Application Protocol) | IoT Testware: CoAP Test Suite | Conformance | Test specifications and TTCN-3 | Protocol conformance: RFC7252 | Eclipse IoT Testware project, https://projects.eclipse.org/proposals/eclipse-iot-testware |
| 6LoWPAN (IPv6 over Low Power Wireless Personal Area Networks) | 6LoWPAN Test Suite | Conformance | Test specifications and TTCN-3 | Protocol conformance: RFC 4944 Transmission of IPv6 over IEEE 802.15.4 RFC 6282 Header compression for 6lowpan | INRIA (French Institute for Research in Computer Science and Automation), Rocquencourt, France, https://www.inria.fr |
| ZigBee [37] | Zigbee Testing and Certification | Conformance | not publicly available | not publicly available | TüV Rheinland, Berlin, Germany, https://www.tuv.com/ and SeaSolve Software, San Jose, CA, United States, http://www.seasolve.com/ |
| LoRaWAN (Long Range Wide Area Network) | LoRaWAN Certification Test Tool (LCTT) | Conformance | not publicly available | Protocol conformance: LoRa Alliance European EU 863-870 MHz Region End Device Certification Requirements document, LoRa Alliance US + Canada US902-928 MHz Region End Device Certification Requirements document | LoRaWAN Alliance, Fremont, CA, United States, https://lora-alliance.org |
| WPAN (Wireless Personal Area Networks) IEEE 802.15.4 | SeaSolve IEEE 802.15.4 | Conformance | not publicly available | not publicly available | SeaSolve Software, San Jose, CA, United States, http://www.seasolve.com/ |
| LDAP (Lightweight Directory Access Protocol) | OpenLDAP tests | Conformance | Linux, CBash-Shell-Scripting | OpenLDAP implementation | OpenLDAP software, https://www.openldap.org/ |

## 8. Towards Reliable Information and Communication Technology for Resilient Smart Cities

This section summarizes our key findings for a viable path towards the establishment of reliable and trustworthy urban ICT for sustainable smart cities across Europe and the globe in the coming decades. We structure the recommendations in four categories (see Table 4) relating to oupPLUS and present calls for action based on the current state of play. The target group for our recommendations is constituted by IT decision makers at city/community level. This includes Chief Digital Officers (CDO) and IT architects of relevant municipal/utility companies in addition to people responsible for tender definition and strategy development towards the establishment of resilient and sustainable ICT-based smart cities. The so defined target group outlines a clear profile of stakeholders within a city development team that take a role between official city representatives, who are normally not familiar with IT terminology, and the core IT technological teams (e.g., within a subcontractor), which on the contrary do not need to be familiar with the final receivers of the provided ICT services.

**Table 4.** oupPLUS-based Recommendations for Resilient and Sustainable Smart Cities.

| Category | Recommendations | oupPLUS Layer/Pillar |
|---|---|---|
| Data Sources | Recommendation 1:<br>Increase deployment of sensors<br>Recommendation 2:<br>Rely on open-source sensor platforms<br>Recommendation 3:<br>Deploy municipal sensor networks<br>Recommendation 4:<br>Diversify the access technologies at the edge<br>Recommendation 5:<br>Place data quality processes close to the data source | 0. Field Equipment/Device Capabilities<br>2. Device Asset Management and Operational Capabilities<br>8. Privacy and Security Capabilities<br>10. Network, Systems and Data Management |
| Network and Connectivity | Recommendation 6:<br>Establish special city network backbones (service-oriented network slices)<br>Recommendation 7:<br>Secure urban ICT by trusted services to avoid hacker attacks<br>Recommendation 8:<br>Establish urban ICT in a redundant manner for more than 99% availability<br>Recommendation 9:<br>Utilize state of the art technologies<br>Recommendation 10:<br>Be open for new technologies and prepare early for upcoming technologies | 1. Communication, Network and Transport Capabilities<br>8. Privacy and Security Capabilities<br>10. Network, Systems and Data Management |
| Solutions | Recommendation 11:<br>Provide catalogues of ready-to-go solutions and components<br>Recommendation 12:<br>Adopt sufficient automated quality assurance measures for urban ICT | 7. Stakeholder Engagement and Collaboration Capabilities<br>9. Common Services Capabilities |
| Processes: | Recommendation 13:<br>Establish certification schemes for smart city solutions<br>Recommendation 14:<br>Make certification facilities continuously available and affordable in order to enable quick recertification<br>Recommendation 15:<br>Urban ICT infrastructure should be managed by agile DevOps like processes, which should also be certified. | 7. Stakeholder Engagement and Collaboration Capabilities<br>8. Privacy and Security Capabilities<br>9. Common Services Capabilities |

## 9. Conclusions

This paper summarizes a series of research studies relating to the systematic development of urban ICT and smart cities, which are based on the ongoing development and standardization activities conducted by Fraunhofer FOKUS. The core topic of this paper is given by the fundamental need to develop reliable and trustworthy ICT solutions and infrastructures for sustainable smart cities. Thereby, we started with a review of various related research and development activities, including long-term initiatives such as FI-WARE or European projects such as Triangulum and Espresso. Afterwards, we laid the fundamentals for further discussions by defining the terms sustainability, smart cities and sustainable smart cities according to which we continued developing a set of relevant concepts on reliability and dependability. In order to systematically approach the topic of resilient and trustworthy urban ICT for sustainable smart cities, we introduced the concept of Open Urban Platforms and corresponding reference architecture models. With oupPLUS, we highlighted the quality assurance requirements for urban ICT, which are central for high quality reliable ICT infrastructure as a backbone for smart city processes. oupPLUS is discussed in detail towards the establishment of sophisticated quality assurance and certification processes for reliable and trustworthy urban ICT. Based on our practical experiences, we provided selected examples for methods and tools (e.g., test suites) and discuss—in relation to the reference architecture—current and upcoming technologies, which are important for smart cities. Based on our lessons learned in numerous practical smart city projects, we provide concrete recommendations for achieving reliable and trustworthy urban ICT and sustainable smart cities within the scope of the oupPLUS Open Urban Platform.

With respect to future work: We are currently establishing further standardization activities within the German DIN Smart City Forum. These activities aim at standardizing the aspects of quality assurance and the various recommendations and related needs, setups and interrelations required to realize the recommendations (from Section 8) and to achieve better sustainable and resilient ICT-based smart cities. Furthermore, Fraunhofer FOKUS—together with industrial partners—has kick-started organizational and technical efforts to establish the discussed oupPLUS-based certification and recertification scheme.

**Author Contributions:** N.T. has made solid contributions during the past years to all related fields discussed in the current publication, especially to the areas and standardization activities for Open Urban Platforms and ICT reference architectures for smart cities. Furthermore, he is an expert in the domain of quality assurance and testing and hence his main contributions are related to Section 6, Section 7 as well as to the recommendations for reliable urban ICT in Section 8. Beyond that, N.T. contributed to shaping all the other sections of the paper. I.S. contributed to this research during her time as team leader and institute director at the Fraunhofer Institute for Open Communication Systems in Berlin. She was intensively involved in the discussion and standardization activities relating to the topics of Open Urban Platforms and design, engineering, and testing support for smart cities. In this regard, her main contributions are reflected in Section 5, Section 7 and especially in the introduction and definitions' parts of this paper, i.e., Sections 1 and 3. All authors have read and agreed to the published version of the manuscript.

**Funding:** This work was partially funded by the projects Triangulum [26], Morgenstadt [35], STREETLIFE [24] and FI-WARE [29].

**Institutional Review Board Statement:** Not applicable.

**Informed Consent Statement:** Not applicable.

**Data Availability Statement:** Not applicable.

**Conflicts of Interest:** The authors declare no conflict of interest.

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
