# Peer review of "Sustainable and Reliable Information and Communication Technology for Resilient Smart Cities"

_smartcities, doi:10.3390/smartcities4010009_

Round 1

Reviewer 1 Report

The aim of the paper is very interesting. However, I understand it is more IT-oriented, I would recommend enlarging the perspective of the target group. If you are formulating the recommendations, you need to specify to whom. The city representatives are not familiar with IT terminology, IT managers should not be familiar with the final receivers of the applied services. This could be written in the summary or in the Limitations chapter. 

From the IT perspective, the paper is very good, but Smart City is not IT only. 

Author Response

>> The aim of the paper is very interesting.
Thank you

>> However, I understand it is more IT-oriented,
>> I would recommend enlarging the perspective of the target group.

See answer of next reviewer's comment.

>> If you are formulating the recommendations,
>> you need to specify to whom.

The following text was added before the table with the recommendations.

"The target group for our recommendations is constituted by IT decision makers
at city/community level. This includes CDO and IT architects of relevant municipal/utility
companies in addition to people responsible for tender definition and strategy development
towards the establishment of resilient and sustainable ICT based Smart Cities."

>> The city representatives are not familiar with IT terminology,
>> IT managers should not be familiar with the final receivers of the applied services.
>> This could be written in the summary or in the
>> Limitations chapter.

The following sentence was added before the recommendations:

"The so defined target group outlines a clear profile of stakeholders within a city
development team that take a role between official city representatives, who are normally
not familiar with IT terminology, and the core IT technological teams (e.g. within a subcontractor),
which on the contrary don’t need to be familiar with the final receivers of the provided ICT services."

>> From the IT perspective, the paper is very good, but Smart City is not IT only.

The following text was added in section 3.2, in order to address the reviewer's comment:

"Based on the above definitions and elucidations, we can see that the term Smart City has a very broad
scope. However, provided that ICT and IT are considered as the efficient boost towards the
development of future Smart Cities in the various areas (health, buildings, citizens’ participation, mobility,
transportation …), the following chapters focus strongly on the IT, telecommunication and data foundations
of an urban infrastructure as a prerequisite for resilient and sustainable Smart Cities and Communities."

Reviewer 2 Report

Abstract

  • Lines 14 to 16: It is evident that ICT in smart cities covers almost all fields of citizens life and their environment. I suggest replacing the examples (which make more sense in the introduction) with a more general phrase (or focusing on something that the article covers).
  • Line 20: I miss a justification sentence that joins the previous concept presented (the importance of the ICT infrastructure) with the need of an open and standardized system, that in the paper is presented as Open Urban Platforms (OUP).
  • Lines 23-24: In my opinion the sentence “whilst reporting on our latest activities and related developments in the research area” suggests that the article is a summary of activities. an article reader might consider this to be a "technical report". My suggestion is to change to a phrase like: "the article presents the activities and developments necessary to achieve a resilient, standardized smart city, based on open platforms".

Introduction

In general, the introduction is well written and is correct. I miss some examples, and the author's definitions with their opinion and point of view about some terms, or concepts, as resilient or sustainable cities, or environments. These terms are described in section 3, but perhaps a mention to the reader about this aspect is necessary.

  • Line 32: I really like the double reference city / community, and the consequent footnote. On some occasions I readed the term “smart environment” but the author's proposal is good.
  • Line 41: There is some consensus that data and information are two distinct entities. It would be interesting to distinguish them with "data and information" or similar. In any case, given the generic nature of the article, it is up to the authors to maintain the duality of terminology.
  • Lines 70-80: The theoretical background of the article makes the objective shown between lines 70 and 80 difficult to follow. I suggest including some "graphical abstract" or figure with a block diagram to help the reader follow the objective through the article.

Related Work

  • Line 96: I miss a sentence stating that European cities and, in detail, German cities have been reviewed. This sentence would justify the opening sentence.
  • Line 98: I suggest changing from “sharing of resources” to “sharing mobility resources”, because the examples are focussed in these concrete resources.
  • Line 125: Because the exposed architectures have been named, but not organized, I think the phrase "above listed reference architectures" might be more suitable "above named reference architectures"

Sustainable Smart Cities

This section is very brief, and could be moved as part of the introduction. In this way, the Related Work section could be supported by the exposed characteristics.

  • Line 167: A brief introduction on how intelligence has been entering the cities is missing. A single sentence for a paragraph gives a very technical look. It is also important to note if the characteristics listed between lines 168 and 182 are a summary of the three references provided or is a contribution from the authors made from these references.
  • Line 183: I miss some examples about each of the features listed below. These examples can be put in each item, although I think it would facilitate the article consistency to put as paragraphs before subsection 2.3.
  • Line 184: I understand that the authors relate the sustainable city to the SDGs. In this part of the article, it would be justified to relate each of the SDGs with characteristics of smart cities. In this way, it can be justified that a sustainable city is one that meets the SDGs.

Reliable ICT

I think the authors justify this section as part of their contribution to the characteristics of smart cities from the ICT point of view. I think the authors justify this section as part of their contribution to the characteristics of smart cities from the ICT point of view. Although the terms are presented, I miss the relationship with the terms of the previous section or with the characteristics presented in section 3. I think that, like the previous section, this section may be before section 2 in which it presents a large number of terms related to technology and which these terms will justify the next section about architectures.

ICT Reference Architectures and Open Urban Platform for Smart Cities

I think this section can be merged with the section “oupPLUS: The Quality Assurance View on Smart Cities”, this is because section 6 is an extension of section 5.

Table 1 is very technical but very large. Contains information that relates section 2 to the architectures description sections. I think the article would be more readable if they are described in tables, or subsections separately.

Technologies of Relevance for oupPLUS

This section describes the technologies that can be used to meet, I understand, the needs of the SAPs presented in the previous section.

From my point of view, something similar happens to table 2 as to table 1, it is very large and it seems more appropriate to present its content in the form of sections.

It is very interesting, especially if the authors relate the technologies in Table 2 with Table 1.

In this section, too, an ontology is presented (Figure 6). I think this figure is a contribution from the authors, they should put if it is original and should be considered a contribution. When presenting the figure as an ontlogy, it would be useful to include a legend about the type of relationship (the arrows) between entities (artifacts) and the arrow labels. As figure 5 is related to figure 4, it would be interesting to be able to organize the artifacts of figure 5 and their relationship with figure 4.

Likewise, figure 5 also seems to be a contribution from the authors, I suggest highlighting this aspect if so.

Table 3 is justified, although it might be useful to adjust it or present the details in text.

Towards Reliable Information and Communication Technology for Resilient Smart Cities

This section constitutes one of the main new contributions. It is justified, since a lot of concepts and technology has been previously reviewed. I miss the relationship of the artifacts of the ontologies presented with the recommendations. I also miss the relationship of the technologies with the recommendations.

These recommendations should be able to answer questions such as at what SAP level of the oupPLUS architecture can the recommendation fit? or what technology of those presented supports the recommendation?

I think that the recommendations should also be related to the criteria that make a sustainable system. That is, the recommendations are made so that an ICT system (which supports a smart city) makes the city more sustainable thanks to the added intelligence.

Summary

Perhaps it would be more scientific to speak of "conclusions". Contributions are correctly described in the text provided. It would be necessary to add future modifications, trends and emphasize more on sustainability.

References

In my opinion there are too many unconventional or unnecessary references for a scientific article. References to dictionaries, such as 2, are best made in the text as a footnote (if the journal allows it) or as a clarification in parentheses in the text itself.

There are also references to technical reports of companies. It is important to look for a publication that is not hosted only on the internet, the ISBN of a book or a conference or the ISSN of a magazine can better ensure the availability of the information source beyond a company website.

Author Response

>> Abstract

>> Lines 14 to 16: It is evident that ICT in smart cities covers almost all

>>fields of citizens life and their environment.

>> I suggest replacing the examples (which make more sense in the

>> introduction) with a more general phrase
>> (or focusing on something that the article covers).

We would like to keep those, since they are important for a Smart City and they are largely ICT based.
Hence, by establishing a resilient and sustainable ICT infrastructure we influence correspondingly
those examples and beyond in a positive way.

>> Line 20: I miss a justification sentence that joins the previous concept presented
>> (the importance of the ICT infrastructure) with the need of an open and standardized system,
>>that in the paper is presented as Open Urban Platforms (OUP).

The following was added arround line 20, in order to address the reviewer's comment:

"A structured way of providing and maintaining an open and resilient ICT backbone for a city/community
is constituted by the concept of an Open Urban Platform. Therefore, ..."

>> Lines 23-24: In my opinion the sentence “whilst reporting on our latest activities and related developments in the research area”
>> suggests that the article is a summary of activities. an article reader might consider this to be a "technical report".
>> My suggestion is to change to a phrase like: "the article presents the activities and developments necessary to achieve a resilient,
>> standardized smart city, based on open platforms".

The last sentence of the abstract was addapted as followis as to address the reviewer's comment:

"Therefore, the current article presents the activities and developments necessary to achieve a resilient,
standardized Smart City, based on Open Urban Platforms (OUP) and the way those serve as a blueprint for each
city/community towards the establishment of a sustainable and resilient ICT backbone."

>>Introduction

>> In general, the introduction is well written and is correct. I miss some examples, and the author's definitions
>> with their opinion and point of view about some terms, or concepts, as resilient or sustainable cities, or environments.
>> These terms are described in section 3, but perhaps a mention to the reader about this aspect is necessary.

A footnote was added to prepare the reader that detailed definitions come in section 3.

"Our specific definitions of sustainability and resiliency in the context of Smart Cities are provided in
section 3 as the article builds up logically."

>> Line 32: I really like the double reference city / community, and the consequent footnote. On some occasions
>> I readed the term “smart environment” but the author's proposal is good.

Thank you

>> Line 41: There is some consensus that data and information are two distinct entities.
>> It would be interesting to distinguish them with "data and information" or similar.
>> In any case, given the generic nature of the article, it is up to the authors to maintain the duality of terminology.

Fixed.

We are using "data and information"

>> Lines 70-80: The theoretical background of the article makes the objective shown between lines 70 and 80
>> difficult to follow. I suggest including some "graphical abstract" or figure with a
>> block diagram to help the reader follow the objective through the article.

A block diagram was added as Figure 1 and the text was adapted to refer to the figure.

>> Related Work

>> Line 96: I miss a sentence stating that European cities and, in detail, German cities have been reviewed.
>> This sentence would justify the opening sentence.

We start the section with the following sentence:

"A large number of Smart City solutions – which we also reviewed as a preparation for the current paper
- have been developed across European and German cities in the past years."

>> Line 98: I suggest changing from “sharing of resources” to “sharing mobility resources”, because the examples
>> are focussed in these concrete resources.
fixed

>> Line 125: Because the exposed architectures have been named, but not organized, I think
>> the phrase "above listed reference architectures" might be more suitable "above named reference architectures"

fixed

>> Sustainable Smart Cities

>> This section is very brief, and could be moved as part of the introduction. In this way,
>> the Related Work section could be supported by the exposed characteristics.

We prefer to keep it as it is.

We added a footnote in the Introduction referring to this section.

>> Line 167: A brief introduction on how intelligence has been entering the cities is missing.
>> A single sentence for a paragraph gives a very technical look.

Following sentence was added:

"Thereby, ICT tools and automations introduce additional intelligence in the city processes, which can be efficiently
utilized towards achieving the SDG s and improving quality of life in cities, communities and across the globe."

>> It is also important to note if the characteristics listed between lines 168 and 182 are
>> a summary of the three references provided or is a contribution from the authors made from these references.

The first sentence in the section was changed to:

"We summarize the Smart City/Community understanding according to to [29],[30], and [31]: "

>> Line 183: I miss some examples about each of the features listed below. These examples can be put in each item,
>> although I think it would facilitate the article consistency to put as paragraphs before subsection 2.3.

Corresponding examples were given in brackets as examples for processes and infrastructures.

>> Line 184: I understand that the authors relate the sustainable city to the SDGs. In this part of the article,
>> it would be justified to relate each of the SDGs with characteristics of smart cities. In this way, it can be justified
>> that a sustainable city is one that meets the SDGs.

This is already done in section 3.1. We added a reference to section 3.1 in the text.

>> Reliable ICT

>> I think the authors justify this section as part of their contribution to the characteristics of smart cities from the
>> ICT point of view. I think the authors justify this section as part of their contribution to the characteristics of
>> smart cities from the ICT point of view. Although the terms are presented, I miss the relationship with the
>> terms of the previous section or with the characteristics presented in section 3. I think that, like the previous section,
>> this section may be before section 2 in which it presents a large number of terms related to technology and which these
>> terms will justify the next section about architectures.
>> ICT Reference Architectures and Open Urban Platform for Smart Cities

We prefer to keep it as it is, since we present the different pieces which are subsequently put together to define
the recommendations and the concepts. This is also reflected in the newly added Figure 1.

>> I think this section can be merged with the section “oupPLUS: The Quality Assurance View on Smart Cities”,
>> this is because section 6 is an extension of section 5.

We prefer to keep it as it is, since we present the different pieces which are subsequently put together to define
the recommendations and the concepts. This is also reflected in the newly added Figure 1.

>> Table 1 is very technical but very large. Contains information that relates section 2 to the architectures description sections.
>> I think the article would be more readable if they are described in tables, or subsections separately.

The table is meant to contain a lot of condensed technical information and examples. The reader can pick the concepts
without going into the details of the table. Hence, we would like to keep these very technical details in the table form.

>> Technologies of Relevance for oupPLUS

>> This section describes the technologies that can be used to meet,
>> I understand, the needs of the SAPs presented in the previous section.

Thanks

>> From my point of view, something similar happens to table 2 as to table 1,
>> it is very large and it seems more appropriate to present its content in the form of sections.

The table is meant to contain a lot of condensed technical information and examples. The reader can pick the concepts
without going into the details of the table. Hence, we would like to keep these very technical details in the table form.

>> It is very interesting, especially if the authors relate the technologies in Table 2 with Table 1.

Thanks.

It is not possible to link table 1 and table 2 in a one-to-one manner. Many of the technoligies span over
multiple layers (of the reference architecture) and SAPs depending on the concrete deployment and solution.

>> In this section, too, an ontology is presented (Figure 6). I think this figure is a
>> contribution from the authors, they should put if it is original and should be considered a contribution.

Yes, this is our contribution.

>> When presenting the figure as an ontlogy, it would be useful to include a legend about the type of relationship
>> (the arrows) between entities (artifacts) and the arrow labels.

We added the following footnote as to refer to the corresponding modelling concepts:

"The used notations originate from the Protégé ontology modelling tool."

>> As figure 5 is related to figure 4, it would be interesting to be able to organize the artifacts of figure 5
>> and their relationship with figure 4.

We have the following text explaining the relationship:

". In Figure 6, we see all artifacts from Figure 5 including the OUP layers, the capability maps for a
solution/component, the SAPs from oupPLUS as well as the abstract interface/protocol stacks for communication
over an SAP as described in Section 6 and exemplified with concrete instances in Table 1."

>> Likewise, figure 5 also seems to be a contribution from the authors, I suggest highlighting this aspect if so.

It is our contribution. In case a figure is adapted and interpreted from the work of others, we always provide
the reference and cite it next to the figure.

>> Table 3 is justified, although it might be useful to adjust it or present the details in text.

The table is meant to contain a lot of condensed technical information and examples. The reader can pick the concepts
without going into the details of the table. Hence, we would like to keep these very technical details in the table form.

>> Towards Reliable Information and Communication Technology for Resilient Smart Cities

>> This section constitutes one of the main new contributions. It is justified,
>> since a lot of concepts and technology has been previously reviewed.

Thank you.

>> I miss the relationship of the artifacts of the ontologies presented with the recommendations.
>> I also miss the relationship of the technologies with the recommendations.

The recommendations are meant to be high level, since they would influence general ICT development strategies.
The mapping to the technologies is omitted, in order to not overload the reader. In an earlier version, we had
such information there but dropped in order to make the text more readable.

>> These recommendations should be able to answer questions such as at what SAP level of the oupPLUS
>> architecture can the recommendation fit? or what technology of those presented supports the recommendation?

Yes, correct.

>> I think that the recommendations should also be related to the criteria that make a sustainable system.
>> That is, the recommendations are made so that an ICT system (which supports a smart city) makes the city
>> more sustainable thanks to the added intelligence.

We already map the recommendations to the pillars/layers of the reference architecture. These pillars/layers
are derived and related (within the corresponding sections) to the sustainable smart city discussions at the beginging.

>> Summary
>> Perhaps it would be more scientific to speak of "conclusions".

Changed

>> Contributions are correctly described in the text provided.

Thanks

>> It would be necessary to add future modifications, trends and emphasize more on sustainability.

The following text was added to the conclusions, in order to address the reviewer's comments:

"With respect to future work: We are currently establishing further standardization activities
within the German DIN Smart City Forum. These activities aim at standardizing the aspects of
quality assurance and the various recommendations and related needs, setups and interrelations
required to realize the recommendations (from section 8) and to achieve better sustainable and
resilient ICT based Smart Cities. Furthermore, Fraunhofer FOKUS together with industrial partners
has kick-started organizational and technical efforts to establish the discussed oupPLUS
based certification and recertification scheme."

>> References

>> In my opinion there are too many unconventional or unnecessary references for a scientific article.
>> References to dictionaries, such as 2, are best made in the text as a footnote
>> (if the journal allows it) or as a clarification in parentheses in the text itself.
>> There are also references to technical reports of companies.
>> It is important to look for a publication that is not hosted only on the internet,
>> the ISBN of a book or a conference or the ISSN of a magazine can better ensure the availability
>>of the information source beyond a company website.

We reworked the references according to the comment.

We kept however some references to companies' papers since they contain some very specific information about
the performance constraints and goals in 4G or 5G mobile networks.

We added additional scientific references.

Reviewer 3 Report

The paper addresses an interesting topic and it is also well designed and quite complete. However, although the authors make an attempt to add some knowledge towards the topic, I have concerns about the contribution of the paper as currently written. Despite the relevance of the topic, after reading the manuscript I see a number of serious problems that I will subsequently outline:

In the introduction I do not understand and see the contribution of the paper as well as the research gap. I think the paper fails to formulate a research problem, which is of interest for the journal community. We do not get answers on what we know now about the topic and what we do not know. The author should more in detail and in a more systematic way present answer on these questions, but also what we need to know. Introduction should be strongly revised accordingly to “industry standards”. Please, motivate your paper and clearly explicated why is this original and how will contribute to theory.

At the end of the intro I also would like to see some preliminary information on the method used and on the results. 

Figure 1 and related discussion may be eliminated (not so high novel value added). 

Literature. The paper seems to exclude from the debate very recent articles on the topic. This is very important because of the novelty of the topic. Also, the position of the paper among key literature is a bit vague, open innovation, KM, smart cities, learning organization. This blurred a bit the boundaries of the research. I am aware that SC context is difficult to position but I would like to ask you to focus more the attention on specific theories and elements.

Moreover, I suggest to include the multistakeholder view of smart city projects, you can find one important stream of literature in many of the paper suggested below:

Ferraris, A., Belyaeva, Z., & Bresciani, S. (2018b). The role of universities in the Smart City innovation: Multistakeholder integration and engagement perspectives. Journal of Business Research.

Another important stream that should be included is (examples):

Appio, F. P., Lima, M., & Paroutis, S. (2018). Understanding Smart Cities: Innovation ecosystems, technological advancements, and societal challenges. Technological Forecasting and Social Change.

Mora, L., Bolici, R., & Deakin, M. (2017). The first two decades of smart-city research: A bibliometric analysis. Journal of Urban Technology, 24(1), 3-27.

Komninos, N., & Mora, L. (2018). Exploring the big picture of smart city research. Scienze Regionali: Italian Journal of Regional Science, 1, 15-38.

Mora, L., Bolici, R., & Deakin, M. (2017). The first two decades of smart-city research: A bibliometric analysis. Journal of Urban Technology, 24(1), 3-27.

Morales-Gualdrón, S. T., & Roig, S. (2005). The new venture decision: An analysis based on the GEM project database. The International Entrepreneurship and Management Journal, 1(4), 479-499.

Method. Not cleary highlighted both in the abstract/intro and within the full paper.

Results. Some figures/tables are uninformative. I would like to see much more novelty as well as specificity in results, please highlight more the more interesting. Sometimes, they are too vague and descriptive and the present form.

Discussion is not so very linear. I suggest to create subsections and to better elaborate on new literatures that you should integrate. Connect much more the paragraphs. Contributions should be better highlighted and elaborated (I would like to see very well developed 3 main contributions to theory). This improved the originality of the paper. Which are the implications? Future lines of research and limitations are too weak.

Author Response

>> The paper addresses an interesting topic and it is also well designed and quite complete.

Thank you

>> However, although the authors make an attempt to add some knowledge towards the topic,
>> I have concerns about the contribution of the paper as currently written.
>> Despite the relevance of the topic, after reading the manuscript I see a number of serious problems that I will subsequently outline:

>> In the introduction I do not understand and see the contribution of the paper as well as the research gap.
>> I think the paper fails to formulate a research problem, which is of interest for the journal community.

We enhanced the introduction of the paper by a figure to clearly show the contributions of the paper in terms
of outputs.

>> We do not get answers on what we know now about the topic and what we do not know.
>> The author should more in detail and in a more systematic way present answer on these questions,
>> but also what we need to know.

The newly added figure 1 contains also the inputs to the paper in terms of concepts which are reviewed
leading to the belonging envisioned outputs of the paper, which are of relevance for the Smart City
community and stakeholders.

>> Introduction should be strongly revised accordingly to “industry standards”.
>> Please, motivate your paper and clearly explicated why is this original and how will contribute to theory.

>> At the end of the intro I also would like to see some preliminary information on the method used and on the results.

The intro was correspondingly adapted and a graphical representation has been added to better exemplify the contribution.

>> Figure 1 and related discussion may be eliminated (not so high novel value added).

Since the other reviewers were positive about the SDG figure, we would like to keep it.

>> Literature. The paper seems to exclude from the debate very recent articles on the topic.
>> This is very important because of the novelty of the topic. Also, the position of the paper
>> among key literature is a bit vague, open innovation, KM, smart cities, learning organization.
>> This blurred a bit the boundaries of the research. I am aware that SC context is difficult to position
>> but I would like to ask you to focus more the attention on specific theories and elements.

We hope that Figure 1 helps to better understand the positioning. There, we clearly define the inputs and theories that influence the discussions and show the outputs of the paper.

>> Moreover, I suggest to include the multistakeholder view of smart city projects, you can find one important
>> stream of literature in many of the paper suggested below:
>> Ferraris, A., Belyaeva, Z., & Bresciani, S. (2018b). The role of universities in the Smart City innovation: Multistakeholder integration and engagement perspectives. Journal of Business Research.
>> Another important stream that should be included is (examples):
>> Appio, F. P., Lima, M., & Paroutis, S. (2018). Understanding Smart Cities: Innovation ecosystems, technological advancements, and societal challenges. Technological Forecasting and Social Change.
>> Mora, L., Bolici, R., & Deakin, M. (2017). The first two decades of smart-city research: A bibliometric analysis. Journal of Urban Technology, 24(1), 3-27.
>> Komninos, N., & Mora, L. (2018). Exploring the big picture of smart city research. Scienze Regionali: Italian Journal of Regional Science, 1, 15-38.
>> Mora, L., Bolici, R., & Deakin, M. (2017). The first two decades of smart-city research: A bibliometric analysis. Journal of Urban Technology, 24(1), 3-27.
>> Morales-Gualdrón, S. T., & Roig, S. (2005). The new venture decision: An analysis based on the GEM project database. The International Entrepreneurship and Management Journal, 1(4), 479-499.

The above literature was integrated in the references and in the state of the art.

>> Method. Not cleary highlighted both in the abstract/intro and within the full paper.

The intro was correspondingly updated, especially by the new figure 1.

>> Results. Some figures/tables are uninformative. I would like to see much more novelty as well as specificity
>> in results, please highlight more the more interesting. Sometimes, they are too vague and descriptive and the present form.

We reviewed the paper and made some minimal adaptations throughout the text trying to approve according to the reviewer's comment.
The newly added figure 1 is basically our main new means in trying to concretely show our contributions.

>> Discussion is not so very linear. I suggest to create subsections and to better elaborate on new
>> literatures that you should integrate. Connect much more the paragraphs.

We reviewed the paper and tried to connect better the paragraphs. Figure 1 (newly added)
also shows the various exisiting inputs to our paper and argues where exactly our contribution lies towards the generation of
valuable contributions.

We also added new sections in the related work as to better structure the arguments and fulfil the requirement of the reviewer.

>> Contributions should be better highlighted and elaborated (I would like to see very well developed
>> 3 main contributions to theory). This improved the originality of the paper.

We refer the reviewer to the newly added figure 1.

>> Which are the implications? Future lines of research and limitations are too weak.

A paragraph on future work was added at the end of the paper

Round 2

Reviewer 2 Report

The authors have done a good job of proofreading. Changes they have not accepted are well justified by the authors. I only have doubts about the very large tables. However, I think this aspect is more related to the requirements of the journal and depends on the editor criteria.

Author Response

Thank you for the valuable feedbacks :-)

Kind regards

Reviewer 3 Report

Dear authors, 

I appreciate your improvements but I do not think that some revisions have not been enough.

For example, arguing that a figure may solve the problem of a lack of theoretical contribution of the paper is not in line with international standard. You have to clearly detail HOW your paper add to a specific literature. 

This is evident in the intro and conclusion section, please revise it accordingly. Also, it seems that you still exclude some relevant studies from the last 5 years. For example, the role of knowledge management and ambidexterity in smart city have been recently addressed in leading journals such as Technological Forecasting and Social Change.

 Good luck with this promising research

Author Response

>> I appreciate your improvements but I do not think that some revisions have not been enough.

>> For example, arguing that a figure may solve the problem of a lack of theoretical contribution of the paper is not in
>> line with international standard. You have to clearly detail HOW your paper add to a specific literature.

>> This is evident in the intro and conclusion section, please revise it accordingly.

We agree with the reviewer. We added the following paragraph describing Figure 1 and our contribution in more detail.

"The overall structure of the paper logic and contribution is provided in Figure 1 with the concept inputs on the left
and derived results for the Smart City community and interested stakeholders on the right. The idea is to take as an
input various aspects from current Smart City theory approaches including blueprints for ICT reference architectures
and belonging standards, various views and dimensions of the sustainability topic, theoretical foundations from the
domain of dependable IT systems, state-of-the-art and emerging technologies as well as theoretical artifacts from
the area of quality assurance and testing for communication based systems. All these inputs are combined within the
paper as to provide an overall picture of a resilient and sustainable ICT based Smart City. This resilient and sustainable Smart
City should build on three main pillars in our view, which are depicted as the papers’ theoretical output on the right
side in Figure 1, namely: the design principles for developing sustainable and resilient ICT for Smart Cities, concrete
recommendations for next steps on technological and organizational level, as well as the concept for continuous quality
assurance and certification processes for the establishment of high quality and secure critical IT and data communication
infrastructures within and urban environment."

>> Also, it seems that you still exclude some relevant studies from the last 5 years.
>> For example, the role of knowledge management and ambidexterity in smart city have been recently addressed in leading journals
>> such as Technological Forecasting and Social Change.

We added the following publications from the mentioned domains and described them briefly in the related work section:

----------------------------------------------------------
Alberto Ferraris, Niclas Erhardt & Stefano Bresciani (2019)
Ambidextrous work in smart city project alliances: unpacking the role of human resource management systems,
The International Journal of Human Resource Management, 30:4, 680-701, DOI: 10.1080/09585192.2017.1291530

Stefano Bresciani, Alberto Ferraris, Manlio Del Giudice,
The management of organizational ambidexterity through alliances in a new context of analysis: Internet of Things (IoT) smart city projects,
Technological Forecasting and Social Change, Volume 136, 2018, Pages 331-338, ISSN 0040-1625,
https://doi.org/10.1016/j.techfore.2017.03.002.

Daniel van den Buuse, Willem van Winden & Wieke Schrama (2020) Balancing Exploration and Exploitation in Sustainable Urban Innovation:
An Ambidexterity Perspective toward Smart Cities, Journal of Urban Technology, DOI: 10.1080/10630732.2020.1835048

John Israilidis, Kayode Odusanya, Muhammad Usman Mazhar,
Exploring knowledge management perspectives in smart city research: A review and future research agenda,
International Journal of Information Management, Volume 56, 2021, 101989, ISSN 0268-4012,
https://doi.org/10.1016/j.ijinfomgt.2019.07.015.
----------------------------------------------------------

We created and extra sub-section "2.2 Knowledge Management for Smart Cities"

"An important related research direction to be considered is given by the topic of knowledge management for Smart Cities
and urban innovation [59]. The idea in this area is to collect and share knowledge regarding different projects, initiatives
and concepts on organizational (e.g. with regard to human resources management [57]) and technical level (e.g. IoT activities [58]),
in order to arrive at a collaborative approach towards the development of future urban environments. Thus, cities are
expected to become knowledge hubs for different participants in an urban innovation eco-system [60]. The involved stakeholders
who are brought on the table for collaboration involve the public and the private sector. This includes city representatives
and departments, citizens, utility companies, SMEs, universities and research institutes as well as large scale industry such
as software power houses or telecom and infrastructure providers."

>> Good luck with this promising research
Thank you very much